# Expanding Genetic Code Expansion through Resource Facilities, Workshops, and Conferences

**DOI:** 10.3390/ijms20092103

**Published:** 2019-04-29

**Authors:** E. James Petersson, Ryan A. Mehl, Christopher A. Ahern

**Affiliations:** 1Department of Chemistry, University of Pennsylvania, 231 South 34th Street, Philadelphia, PA 19104-6323, USA; 2Department of Biochemistry and Biophysics, Oregon State University, 2011 Ag Life Sciences Building, Corvallis, OR 97331, USA; 3Department of Molecular Physiology and Biophysics, The University of Iowa, 4270 Carver Biomedical Research Building, 285 Newton Road, Iowa City, IA 52242, USA

**Keywords:** genetic code expansion, unnatural amino acid, non-canonical amino acid, amber suppression

## Abstract

Genetic Code Expansion (GCE) enables the encoding of amino acids with diverse chemical properties. This approach has tremendous potential to advance biological discoveries in basic research, medical, and industrial settings. Given the multiple technical approaches and the associated research activities used to achieve GCE, herein we have taken the opportunity to describe ongoing out-reach efforts in the GCE community. These include Resource Facilities that nucleate expertise and reagents within a specific GCE discipline, hands-on Workshops to provide GCE training, and GCE Conferences which bring the community together in a collegial setting. The overall goal of these activities is to accelerate the integration of GCE approaches into more research settings and to facilitate solutions to common technical hurdles.

The field of Genetic Code Expansion (GCE) has the potential to profoundly affect a broad range of fields, from in vitro biophysics to cell and animal biology, and even non-biological fields like materials science. The ability to impact these areas ultimately depends on the dissemination of robust, consistent GCE components by developer laboratories and on the adoption of GCE methods by non-developer laboratories. To these ends, several international efforts have been put forth in recent years in the forms of resource facilities, workshops, and the first two GCE conferences. A resource facility is one that provides reagents and expertise to non-developer laboratories to enable their use of GCE methods, even for single sets of experiments, so that they can capitalize on GCE tools without being required to become experts in the more nuanced aspects of GCE. Workshops provide hands-on training to those who wish to establish GCE methods in their own laboratory and plan to use GCE in the longer term, perhaps ultimately becoming developer laboratories themselves. Conferences, of course, facilitate sharing of new research ideas and forming relationships that may lead to collaborations. This is particularly valuable for non-developer groups who wish to understand the scope of tools that could be applied to their area of interest, but also to developer laboratories wishing to discuss best practices at a level of technical detail too great for more general audiences at typical Chemical Biology conferences. Here, we highlight recent examples in each of these categories to demonstrate their value and raise awareness for future endeavors.

GCE, as a method, enables researchers to add an almost limitless number of new types of amino acid chemical functionality to a protein, in a site-directed manner and within a living cell. The tremendous potential for this method is counter-balanced by the potential caveats in its application to each amino acid, specific protein, or scientific discipline. Therefore, broadening the role of GCE is the core mission of the Unnatural Protein Facility, which was started by Prof. Ryan Mehl of the Oregon State University Biochemistry and Biophysics Department in 2012 to give researchers full access to current GCE tools and to facilitate development of new tools desired by the community. The facility has hosted GCE workshops annually since 2015, with the main goal of educating scientists on how to optimize GCE tools and successfully use GCE in their research. “The exponential growth of the field is both a blessing and a curse for educating new scientists in GCE, forcing the Facility to add newly available methods each year to the GCE workshop,” says Ryan Mehl. The success of the GCE workshop can be attributed to the hands-on lab training focus using the attendees’ proteins and amino acids of interest and the goal that attendees leave with the knowledge to train other lab members in successful GCE experimental design. Since many newcomers to the field believe that GCE can be applied to all organisms and all proteins easily using hundreds of different amino acid functionalities, researchers often arrive at the facility with an extremely broad set of questions and problems. Each year the Facility works with GCE developers to gather the optimal GCE tools and conditions to help attendees succeed with their specific problem of interest.

The laboratory component focuses on two common needs in GCE: (1) incorporation of an existing non-canonical amino acid into a protein of interest, and (2) screening a new non-canonical amino acid against a panel of synthetases to determine if the desired non-canonical amino acid can be incorporated with current GCE tools. As part of the laboratory component of the course, participants are given the opportunity to bring their research gene of interest and/or their own non-canonical amino acid to the workshop to test expression and incorporation along with optimized controls. As the capabilities of GCE have grown, the Facility has added lab components on expression of unnatural proteins in mammalian systems. To ensure long term success with GCE for the participants, the workshop lectures cover many critical areas: (1) how new GCE components are generated; (2) how attributes of orthogonality, efficiency, fidelity and permissivity are defined, measured, and conditionally dependent; (3) the major applications of new non-canonical amino acids, and (4) the future of GCE and its limitations.

The GCE workshop attendance has been limited to 20 scientists, ensuring that each attendee obtains the knowledge needed to succeed in GCE. As major global companies and more diverse scientists require training each year, the Facility plans to add additional workshops to meet the demands of the growing community. This growth also inspired convening of the first dedicated GCE conferences, as it became clear that the critical mass for such an event was present in the field and in new users entering the field.

The first and second GCE conferences, held in August 2016 and 2018 on the Oregon State University campus, brought together GCE method developers and users to share their latest scientific discoveries as well as make plans to increase the accessibility of the methods and diversity of the field. The GCE conferences were held in conjunction with the second (2016) and fourth (2018) GCE workshops. In 2018, more than 100 scientists from 12 countries attended the workshop and conference. One of the key themes of the conferences was expanding access to GCE methods and the scope of experiments for which one envisions applying GCE techniques. The 2016 conference was hosted by Ryan Mehl and the Unnatural Protein Facility staff and co-organized by Prof. Jason Chin of the Medical Research Council and Cambridge University. The 2018 conference was co-organized by Prof. E. James Petersson of the University of Pennsylvania and Prof. Carsten Schultz, who holds appointments at the European Molecular Biology Laboratory (EMBL) in Germany and Oregon Health and Science University. Having co-chairs from multiple continents helped to foster international participation to gain a broad perspective on the field and provide the potential to disseminate best practices worldwide.

The 2018 keynote speakers highlight the scope of researchers at these conferences. Jason Chin is one of the leading developers of new GCE methods, one who not only provides researchers with new synthetases, but thinks broadly about reprogramming translation as a whole. Prof. Thomas Sakmar of the Rockefeller University has used GCE methods longitudinally to study membrane protein signaling through spectroscopic methods and in drug discovery. Other speakers at each conference included both GCE developers and users, and session topics ranged from fundamental studies of protein function and drug binding interactions to the engineering of viruses for therapeutic studies. Listings of the full range of 2016 and 2018 speakers can be found at the GCE Conference website: gce-conference.org. Since there was broad enthusiasm for holding another GCE Conference, and agreement that a non-U.S. location would be beneficial to the GCE community, Peng Chen will to host the 2020 meeting at Peking University. He will be assisted by co-chair Edward Lemke of EMBL and chairs-in-waiting Abhishek Chatterjee of Boston College and Kathrin Lang of Technical University of Munich.

Additional efforts to disseminate GCE technology are underway in the Ahern lab at the University of Iowa through a community resource termed “The Facility for Atomic Mutagenesis”. The goal of these efforts is to assist client labs in the application of chemically acylated tRNA for GCE. Data were presented at the GCE conference by facility director, Prof. Chris Ahern, which highlighted the utility of acylated tRNA in the study of ion channel proteins in the *Xenopus laevis* oocyte expression system. The current approach utilizes synthetic pyrrolysine (tRNAPyl) tRNA for encoding subtle variants of amino acids. This improvement takes advantage of recent improvements in long synthetic RNA (Integrated DNA Technologies, Iowa City, IA, USA). A strength of the acylated tRNA approach is that a single tRNA type, such as tRNAPyl, can be used to encode many types of chemically diverse amino acid substrates. Additionally, very little of the amino acid, often in the 10–30 mg scale, is required to obtain significant amounts of data. This is because application in the (1 µL) oocyte requires small amounts of injected acylated tRNA. Further, the approach can be useful when variations are modest, sometimes representing single atom changes to an amino acid—cases where it may be challenging to identify an evolved synthetase. At the University of Iowa, the tRNA resource also allows users to request custom chemistry projects to suit the scientific needs of the desired experimental outcomes. Pursuant to this goal, the facility and its users strive to tighten the gap between high-resolution structural discoveries, enabled by the surge in transmission electron cryo-microscopy (Cryo-EM) applications, and ever more powerful computational efforts. On-site training in the application of these regents is available on a per user basis. These efforts are currently supported by an NIH/NINDS R24 award. For more information on the Facility can be found at www.ahernlab.com.

Combining resources and workshops with broad dissemination platforms like plasmid distribution services and commercial vendors for amino acids will further grow GCE. This is not just true for biology labs that may not have the expertise to synthesize amino acids, but also for chemists who have the expertise but don’t wish to dedicate the person-power for the sake of a one-off experiment. At the 2018 GCE Conference, Figure 1, an afternoon forum was held with Addgene and Bachem to discuss the best ways of disseminating plasmids and novel amino acids, as well as facilitating GCE usage in academia and industry. The Addgene GCE page, which provides more than 50 synthetase plasmids, can be found at addgene.org/genetic-code-expansion. The most broadly used amino acids are now commercially available, and efforts to distribute others systematically are underway.

## Conclusions

These examples highlight the value of resource facilities, workshops, and conferences to contribute to the expansion of GCE methods and applications. They are not presented to be definitive examples, but to raise awareness and provide ideas for similar efforts that could further grow the field. To this end, the three co-authors welcome correspondence about their respective endeavors (R24 resource, Ahern; 2016 GCE Conference and 2014–2018 workshops, Mehl; 2018 GCE Conference, Petersson). There are certainly technical challenges yet to be met, as one still hears questions about whether GCE “really works” despite its many successes. There is also a need to expand the diversity of the field, in terms of the scientific backgrounds of users, but also in terms of their ethnicities and genders. Greater inclusiveness of both types will provide new perspectives and help to broaden the scope of GCE research. The impact of GCE is only beginning to be felt, and efforts such as these can help to more fully realize its potential.

## Figures and Tables

**Figure 1 ijms-20-02103-f001:**
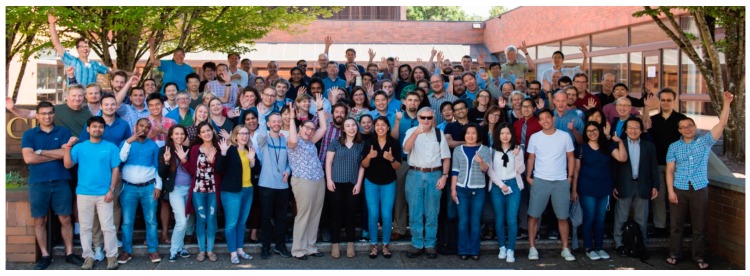
The 2018 GCE Conference at Oregon State University was attended by more than 100 academic and industrial scientists from 12 countries (Photo by Hanna O’Leary from OSU Media Services).

