# Peer review of "Expanding Genetic Code Expansion through Resource Facilities, Workshops, and Conferences"

_ijms, 2019, doi:10.3390/ijms20092103_

Round 1

Reviewer 1 Report

Technologies of the genetic code expansion (GCE) has the potential to contribute not only life science but non-biology field. To be widely disseminated this technology, some efforts such as establishment of resource facility and holding workshops and conferences has been started by Prof. Ryan Mehl of the Oregon state University. This report from Petersson et al tell us the recent and future activities of the GCE society and provides websites in terms of workshop/conference and resource for GCE experiments. I think that this report would help to disperse their efforts about GCE technology and there are no major issues that would need to be addressed prior to publication.

Minor points

L. 93 “pyrrolysyl (Pyl) tRNA” be pyrrolysine tRNA (tRNAPyl)

L. 96. “PylT” should be tRNAPyl

Author Response

We thank the reviewer for their time and positive comments.  We have made the proposed corrections to the manuscript.  

Reviewer 2 Report

This report  nicely described the objective, organization, approach, and perspective for the Genetic Code Expansion Conference. It should be a nice advertisement for attracting more and more scientists into this field.

Author Response

We thank the reviewer for their time and positive comments. 

Reviewer 3 Report

For those not familiar with the concept of genetic code expansion (GCE) I think an explanatory sentence or two at the start of the report would be extremely helpful. This could include brief mention of the sorts of techniques involved and the actual and potential applications of the field.

Author Response

We thank the reviewer for the suggestion to add some context for the GCE method.  We agree that this may help to broaden the impact of readers of the journal.  

We have now added the following two sentences to the first paragraph of the manuscript.

"GCE as a method enables researchers to add an almost limitless number of new types of amino acid chemical functionality  to a protein, in a site-directed manner and within a living cell.  The tremendous potential for this method is counter-balanced by potential caveats in its application to each amino acid, specific protein, or scientific discipline.   ..."

Reviewer 4 Report

This paper is a conference report of some recent workshops and conferences which focus on the genetic code expansion. The content of the paper is mainly organized from the aspect of dissemination of the genetic code expansion technology from developer laboratories to non-developer laboratories. This paper is not only a conference report but also a useful manual, including how to obtain experimental and informational resources, for researchers who are willing to use the genetic code expansion technology. The presentation of this paper is attractive and reader-friendly.

Author Response

(The authors gave the same response as above.)
